# Fighting Fatigue in Systemic Lupus Erythematosus: Experience of Dehydroepiandrosterone on Clinical Parameters and Patient-Reported Outcomes

**DOI:** 10.3390/jcm11185300

**Published:** 2022-09-08

**Authors:** Oliver Skoglund, Tomas Walhelm, Ingrid Thyberg, Per Eriksson, Christopher Sjöwall

**Affiliations:** Department of Biomedical and Clinical Sciences, Division of Inflammation and Infection/Rheumatology, Linköping University, SE-581 85 Linköping, Sweden

**Keywords:** dehydroepiandrosterone, systemic lupus erythematosus, patient-reported outcomes, fatigue, SLEDAI-2K

## Abstract

Manifestations related to ongoing inflammation in systemic lupus erythematosus (SLE) are often adequately managed, but patient-reported outcome measures (PROMs) support that fatigue and low quality of life (QoL) in the absence of raised disease activity remain major burdens. The adrenal hormone dehydroepiandrosterone (DHEA) has shown potential as a pharmacological agent for managing fatigue in mild SLE. We retrospectively evaluated data on dosage, disease activity, corticosteroid doses, concomitant antirheumatic drugs, and PROMs regarding pain intensity, fatigue, and well-being (visual analogue scales), QoL (EQ-5D-3L) and functional disability. A total of 15 patients with SLE were exposed to DHEA and 15 sex- and age-matched non-exposed SLE patients served as comparators. At baseline, 83% of the DHEA-exposed patients had subnormal DHEA concentration. The 15 subjects prescribed DHEA were exposed during a median time of 12 months (IQR 16.5) [range 3–81] and used a median daily dose of 50 mg of DHEA (IQR 25.0) [range 25–200]. Neither disease activity, nor damage accrual, changed significantly over time among patients using DHEA, and no severe adverse events were observed. Numerical improvements of all evaluated PROMs were seen in the DHEA-treated group, but none reached statistical significance. For DHEA-exposed patients, a non-significant trend was found regarding fatigue comparing baseline and 36 months (*p* = 0.068). In relation to SLE controls, the DHEA-exposed group initially reported significantly worse fatigue, pain, and well-being, but the differences diminished over time. In conclusion, DHEA was safe, but evidence for efficacy of DHEA supplementation in relation to PROMs were not found. Still, certain individuals with mild SLE, plagued by fatigue and absence of increased disease activity, appear to benefit from DHEA in terms of improved fatigue and QoL. Testing of DHEA concentration in blood should be performed before initiation, and investigation of other conditions, or reasons responsible for fatigue, must always be considered first.

## 1. Introduction

Systemic lupus erythematosus (SLE) is a systemic autoimmune condition with potential to affect virtually any of organ system. The female-to-male ratio of patients with SLE is approximately 9:1 and most cases are diagnosed between 15 and 44 years of age [1,2]. The pathogenesis of SLE remains to be fully uncovered, but is a product of a complex interplay between hereditary and environmental factors, such as ultraviolet light exposure, certain infections, and drugs, leading to dysfunctional disposal of cellular debris [1,2]. Periods of raised disease activity may be followed by longtime remission and the disease severity ranges from mild skin and joint manifestations to life-threatening cytopenia and central nervous system (CNS) disease [2].

While the number of treatment options for SLE steadily have increased, health related quality of life (QoL) and fatigue remain major burdens for patients in their everyday living. In a recent review article, based on data from 570 patients with SLE, 68% reported fatigue and 37% severe fatigue [3]. In an older systematic review, involving 9886 cases, it was shown that 34% (95% confidence interval [CI] 24–44%) had some form of work disability related directly or indirectly to their disease [4]. Data from Sweden confirm that the indirect costs for SLE are substantial [5].

The cause of fatigue experienced by patients with SLE is likely to be multifactorial [6,7]. An association between increased disease activity and fatigue exists, but fatigue is often present even in the absence of any detectable SLE activity [8,9]. Dehydroepiandrosterone (DHEA), derived from cholesterol, has achieved attention as a potential candidate to reduce fatigue and mild disease activity in certain patients with SLE [10]. The motive to use DHEA in SLE is strengthened by several aspects. Firstly, in animal studies, supplementation of DHEA has shown clear anti-inflammatory effects on the immune system and beneficial effects in lupus-prone mice have been observed [11,12,13]. Secondly, DHEA plasma concentrations are subnormal in a subset of subjects with SLE [14,15]. Thirdly, some 20 years ago, DHEA was evaluated in randomized controlled trials (RCTs) with encouraging results as a potential pharmacological agent in the treatment of mild SLE [16,17,18,19].

At our university unit, we have approximately 20 years’ experience of DHEA as rescue therapy for severe fatigue in mild SLE where other pharmaceutical and non-pharmaceutical interventions were unsuccessful. Herein, we systematically evaluated our retrospective DHEA data in SLE in relation to tolerance, dosage, affected organ systems, disease activity measures, corticosteroid use, concomitant immunosuppressive therapies, and patient-reported outcome measures (PROMs). Sex- and age-matched SLE patients, unexposed to DHEA, served as controls. In addition, as all DHEA-exposed SLE patients had joint/musculoskeletal involvement, we included a second comparator group of patients with early rheumatoid arthritis (RA).

## 2. Methods

### 2.1. Data Source, Patients, and Study Design

This study was a retrospective unblinded observational study including 15 patients with SLE prescribed daily DHEA in various doses under careful follow-up. All patients with SLE were part of the research and quality register *Clinical Lupus Register in North-eastern Gothia* (Swedish acronym KLURING) at Linköping University Hospital, a tertiary referral center with a long experience of management of SLE [20]. These 15 patients represent all patients with SLE exposed to DHEA within the catchment area of Linköping healthcare district since the year 2000. As a control population, another 15 selected subjects with SLE from KLURING, living in the same geographical area but unexposed to DHEA, were included and subsequently age- and sex-matched to the group exposed to DHEA.

As an additional comparator group, 45 patients with early RA from the *2nd Timely Intervention in Early Rheumatoid Arthritis* (Swedish acronym TIRA-2) at Linköping University Hospital [21], living in the same geographical area, were included and matched 3:1 to each participant in the group of patients with SLE exposed to DHEA (Table 1).

Retrospective patient data were retrieved from March 2002 to March 2022 for the three groups based on physical visits to the Rheumatology unit, Linköping University Hospital. The patients were followed up to 36 months with (at least) annual visits. Inclusion criteria for the two SLE groups were age ≥18 years and fulfillment of the 1982 American College of Rheumatology (ACR) and/or the 2012 Systemic Lupus International Collaborating Clinics (SLICC) classification criteria [22,23]. In the DHEA-exposed SLE group, 2/15 fulfilled the SLICC criteria in the absence of meeting ACR criteria; and among SLE comparators, 3/15 fulfilled the SLICC criteria in the absence of meeting ACR criteria. Data from the TIRA-2 cohort was collected 2006–2011 and inclusion criteria were symptom duration (defined as first observed joint swelling <12 months), and either fulfilment of the 1987 American Rheumatism Association criteria or suffering from morning stiffness >60 min, symmetrical arthritis, and small joint engagement [21,24]. As reflected in Table 1, the indication for DHEA supplementation was unmanageable fatigue where other pharmaceutical and non-pharmaceutical interventions had been unsuccessful. In most cases, DHEA concentration in plasma (measured by electrochemiluminescence immunoassay) before treatment initiation was available. Since reference intervals for DHEA are dependent on age and sex, the percentage of the lower reference limit for each included patient was provided.

### 2.2. Assessments

We assessed PROMs at month 0 (baseline), 12, 24, and 36. The included PROMs were collected by questionnaires, including the Swedish version of Health Assessment Questionnaire (HAQ) to assess functional disability (0 = no disability, 3 = severe disability) [25]. The Euro-QoL 5 dimensions (EQ-5D-3L) was used to assess general health based on five different dimensions (mobility, self-care, usual activities, pain/discomfort and anxiety/depression) in order to derive a utility index which provides a score indexed at 1 (perfect health) and 0 (dead) [26] and the visual analogue scale (VAS) of pain, fatigue, and well-being (0–100; 0 = no symptoms, 100 = worst imaginable symptoms), that patients completed at every visit to the Rheumatology unit [27].

SLE disease activity was assessed by the SLE disease activity index-2000 (SLEDAI-2K) and the physician’s global assessment (PGA, graded 0–4; 0 = remission, 4 = maximum disease activity), irreversible organ damage was assessed by the SLICC/ACR damage index (SDI) [28,29]. 

### 2.3. Laboratory Analyses

Longitudinal blood samples were evaluated to detect effects, or any side-effects related to laboratory variables. Hemoglobin concentration, blood cell counts, estimated glomerular filtration rate (eGFR) according to the MDRD 4-variable equation [30], erythrocyte sedimentation rate (ESR), C-reactive protein (CRP), creatine kinase (CK), complement protein 3 (C3) and 4 (C4) were available. Among patients with early RA, only ESR and CRP were available.

### 2.4. Statistics

The groups were compared using the Kruskal–Wallis test to determine any significance between the three groups regarding PROMs, baseline characteristics and for laboratory values where appropriate. The Mann–Whitney *U* test was used to confirm any significance between two of the groups. Spearman’s rho was applied to measure the strength of association between two variables. No adjustments of uncensored data were made. For comparison regarding number of fulfilled ACR criteria between the SLE groups, χ² testing was used. In addition, the group exposed to DHEA was examined using the Wilcoxon signed-rank test to study changes over time in comparison with baseline values. Finally, descriptive statistics were used to display patient characteristics, PROMs, and laboratory values. Statistical analyses were performed using the SPSS software version 28.0.0.0 (SPSS Inc., Chicago, IL, USA) and Prism 9.3.1 (GraphPad Software Inc., La Jolla, CA, USA) for construction of graphs.

### 2.5. Ethics Approvals

Oral and written informed consents were obtained from all patients. The study was conducted according to the Declaration of Helsinki and approved by the Regional Ethics Boards regarding SLE (Linköping M75–08/2008) and early RA (Linköping M168–05).

## 3. Results

### 3.1. Baseline Differences between Patient Groups

The three patient groups did not significantly differ in sex, age at baseline (start of follow-up), or age at onset of rheumatic disease (Table 1). Neither were ethnicity, BMI, SLEDAI-2K scores, PGA, steroid dosage, or disease phenotypes (fulfilled ACR criteria) at baseline different between the two groups of SLE patients. However, the disease duration among patients with early RA was significantly shorter (*p* < 0.001) compared to the DHEA-exposed SLE group.

### 3.2. Disease Activity and Organ Damage

Accrual of organ damage, assessed by SDI, was not different between the two SLE groups (*p* = 0.65) at baseline, nor at the 36-month follow-up (*p* = 0.46). SDI did not change significantly over the 36 months among patients exposed to DHEA (*p* = 1.0; not shown). Global disease activity, assessed by the SLEDAI-2K, was unchanged over time (*p* = 0.32) (Figure 1). Similarly, PGA did not change significantly over time and no severe flares were observed.

### 3.3. Background Medication

At baseline, 12 of 15 patients treated with DHEA used hydroxychloroquine (HCQ) compared to 11 of 15 in the SLE group unexposed to DHEA. No other obvious differences between the groups in use of other disease-modifying antirheumatic drugs (DMARDs) were seen. The median daily dose of prednisolone at DHEA initiation was 5 mg (interquartile range [IQR] 3.75) compared to 2.5 mg (IQR 5) in the SLE controls (*p* = 0.87). As shown in Table 2, three of the DHEA-exposed patients were able to reduce their dose of prednisolone during the study period. Furthermore, among the SLE controls, 3 of 15 reduced the prednisolone dose during follow-up.

### 3.4. DHEA Exposure and Safety

Of the 15 patients exposed to DHEA, 2 (13%) had DHEA concentrations within reference intervals, 10 (67%) showed plasma levels below the lower reference limit and in 3 (20%) cases DHEA had not been analyzed at baseline. In all individuals where a second assessment of DHEA concentration was performed (i.e., after initiation of DHEA), the levels had increased to concentrations within, or even above, the age- and sex-specific reference limits.

The 15 subjects treated with DHEA were exposed for a median of 12 months (IQR 16.5) [range 3–81] and used a median daily dose of 50 mg of DHEA (IQR 25.0) [range 25–200]. As shown in Table 2, DHEA treatment with no major adverse events were observed but 9/15 ceased DHEA therapy during the 36 months. Three patients (20%) had early cessations (≤4 months) due to lack of efficacy or androgenic side effects (acne). Later terminations were usually related to lack of efficacy rather than to side-effects, which mainly were of androgenic nature and deemed as mild (Table 2). Two patients remained on DHEA much longer than the 36-month follow-up and were monitored regularly as part of clinical routine, at least annually.

### 3.5. Longitudinal Effects on PROMs among DHEA-Treated Patients

PROMs at the 12-, 24- and 36-month follow-up for the exposed group were compared with respect to the baseline values (Figure 2A–E). In the DHEA-treated SLE group, numerical improvements of all evaluated PROMs were seen but none of them reached statistical significance over 36 months. A comparison of VAS fatigue between baseline and 36 months yielded a non-significant trend (*p* = 0.068). VAS fatigue at baseline did not correlate significantly with DHEA either expressed as percentage of lower reference limit (Spearman’s rho = 0.078, *p* = 0.82) or as µmol/L (Spearman’s rho = 0.312, *p* = 0.35). The response to DHEA was not different among patients fulfilling the ACR criteria and those who met the SLICC criteria only.

### 3.6. Effects on PROMs between the Patient Groups

Prior to DHEA supplementation (baseline), the DHEA-exposed group reported significantly worse fatigue, pain, well-being, and QoL compared to the unexposed SLE group (Figure 2A–D), but the differences diminished over time. In contrast, the functional disability was worse in the RA group compared to the other two groups (Figure 2E).

### 3.7. Effects on Laboratory Variables

Data on ESR and CRP were available for comparison in all patient groups. ESR was higher in patients with RA at baseline but, in contrast to CRP, the significance diminished over time (Figure 3A,B). C3, C4, hemoglobin concentration, and leukocyte count remained stable over time in the two SLE groups (Figure 3C–F). Although the data indicate a slight worsening of eGFR over time in both SLE groups, no significant differences in eGFR at baseline (*p* = 0.12) or at the 36-month follow-up (*p* = 0.41) were observed (Figure 3G). Further analyses of platelet, neutrophil, and lymphocyte counts as well as CK showed no significant changes over time.

## 4. Discussion

Treatment options for fatigue are limited and remain an unmet need for many patients with SLE, and studies evaluating interventions for fatigue isolated from raised disease activity are rare. The scientific evidence of using DHEA for severe fatigue in mild SLE remains limited. However, albeit small, this retrospective observational unblinded study includes longitudinal follow-up data of a well-characterized DHEA-treated population in a real-life clinical setting and complements previously published RCTs [16,17,18,19].

In contrast to most studies, we herein primarily investigated improvement of PROMs. Pain, fatigue, wellbeing, QoL and functional disability are repeatedly ranked by patients as very important parameters [6]. Unfortunately, according to our data, the effects of DHEA on PROMs on a group level were mediocre or absent. This does not exclude that certain individuals could still benefit from DHEA treatment. Four of fifteen patients had been using DHEA for ≥30 months at the study’s last follow-up. We further show that supplementation of DHEA to patients with SLE is generally safe. Mild side-effects were seen and some had an early cessation, but no severe adverse events were observed. Our patients had mild SLE with low disease activity at baseline, and no severe flares were seen during follow-up. Reassuring was that the DHEA-treated group did not accumulate more organ damage than their unexposed controls.

Originally, the idea to use DHEA in lupus arose from animal studies. In lupus-prone mice (NZB/W F1), administration of DHEA at 2 months of age significantly prolonged survival in exposed animals [12]; at 41 weeks, 71% of the DHEA-treated mice were alive compared to 22% of controls (*p* = 0.04). Moreover, DHEA injections (beginning at 2 months of age) delayed the formation of anti-double-stranded (ds) DNA antibodies in 62% of the treated animals although the antibody levels eventually rose regardless of treatment [12]. In a similar study, NZB/W mice were given DHEA and compared to unexposed controls [11]. The survival between the control and treatment group differed greatly at 12 months, with 64% survival in the DHEA-treated group versus 17% in the control group. In addition, the formation of anti-dsDNA antibodies was halted and remained comparably low in the DHEA-treated group [11]. Finally, immunofluorescence of renal tissue from controls and exposed mice at 6 months showed that DHEA-treated mice had less deposits of immunoglobulin complexes in the kidney, indicating less severe disease progression [11].

Studies evaluating DHEA in patients with SLE show mixed results [16,17]. In a multicenter double blinded RCT, 120 female patients with mild to moderate SLE were given 200 mg of DHEA or placebo over 6 months [18]. Disease activity, assessed by the Systemic Lupus Activity Measure (SLAM) and SLEDAI, [28] showed no statistical difference between the groups after 6 months, but the patient’s global assessment scale in the DHEA-treated group were significantly better than placebo (*p* = 0.005). Moreover, fewer disease flares were noted over the 6-month period in the treatment group (18.3% vs. 33.9%, *p* = 0.010) [18]. In another RCT, including 381 cases with SLE over 27 centers in the United States, patients were randomized to placebo or 200 mg of DHEA for 52 weeks [19]. In patients with active SLE at baseline (defined as SLEDAI > 2), a significant improvement in SLAM (*p* = 0.017) was observed. Moreover, significantly more patients receiving placebo noted a worsening of the patient’s global assessment compared to patients given DHEA (10.9% of DHEA group versus 22.6% in the placebo group, *p* = 0.007). The authors concluded that 200 mg of DHEA can improve and stabilize SLE disease activity in women with mild to moderate SLE and is generally well tolerated [19].

In a small Swedish study, lower doses of DHEA (20–30 mg daily) were investigated. The first 6 months of the study was blinded, and the latter 6 months open-label, in which all patients received DHEA [31]. DHEA was given to 20 patients and 17 received placebo. Physical and mental self-rated QoL was evaluated after 6 and 12 months of treatment. At the 6-month follow-up, the DHEA-treated group reported significant improvements in physical and emotional self-rated health in the questionnaire SF-36 compared to placebo (*p* < 0.05). Despite the small sample size, an observation was made that women with DHEA within reference limits at baseline showed similar improvements in the questionnaires as those with low DHEA. Overall, the results were less clear during the open-label phase and the authors concluded that for some patients, a lower dose of DHEA may be enough to improve well-being and QoL [31].

Our study has several limitations. It was not an RCT, which must be considered. The retrospective observational nature of the data inevitably leads to selection bias. Patients who experienced beneficial effects of DHEA were likely to continue, and thus reporting improved PROMs, compared to those who ceased and were excluded from the analysis. The included study population, with only 15 subjects exposed to DHEA, limits the statistical power and possibility of detecting significant and meaningful differences. The fact that dropouts were higher among DHEA-exposed patients compared to comparators, unequivocally leading to uncensored data, was a major limitation. In addition, all DHEA-treated patients had mild SLE without significantly raised disease activity at baseline; this makes it impossible to evaluate any effects of DHEA on SLE activity. Furthermore, functional disability assessed by HAQ may not be relevant to all patients with SLE and, although it has been used in SLE, HAQ is only validated for RA [32]. Androgenic side-effects were indeed seen in some patients, but no systematic assessment of other gonadal hormones than DHEA was performed. However, none of the patients with hypothyroidism and diabetes had to adjust their doses of levothyroxine or insulin during DHEA exposure. In contrast, major strengths of the study include the Swedish healthcare system’s universal access as well as the long experience of treating patients with SLE at one tertiary referral center and longitudinal follow-up by a limited number of experienced rheumatologists. In addition, we included data from relevant SLE and RA comparators, who experienced similar clinical manifestations as the patients exposed to DHEA, living in the same geographical region of Sweden.

## 5. Conclusions

To conclude, this observational study, including longtime real-life use of DHEA in SLE, is one of very few to date. No serious adverse events were observed, but generally we did not find support for efficacy of DHEA supplementation on PROMs. Still, some individuals with mild SLE, plagued by fatigue and absence of increased disease activity, may obviously benefit from DHEA supplementation in terms of improved fatigue. Testing of DHEA concentration in blood should be performed before initiation, and investigation of other conditions, or reasons responsible for fatigue, must always be considered first.

## Figures and Tables

**Figure 1 jcm-11-05300-f001:**
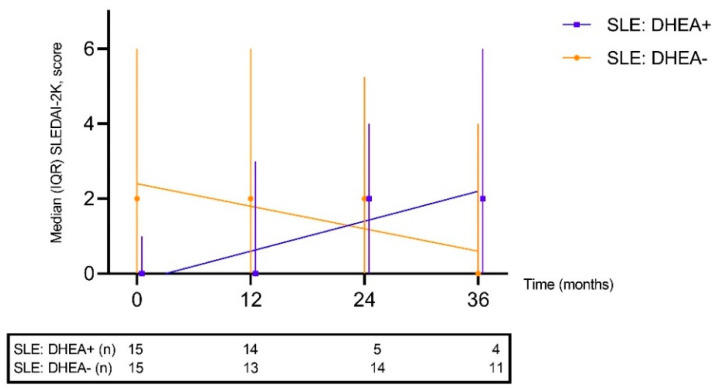
Global disease activity assessed by the systemic lupus erythematosus disease activity index-2000 (SLEDAI-2K) over time depicted for patients exposed to DHEA and for sex- and age-matched controls. No significant changes over time were observed.

**Figure 2 jcm-11-05300-f002:**
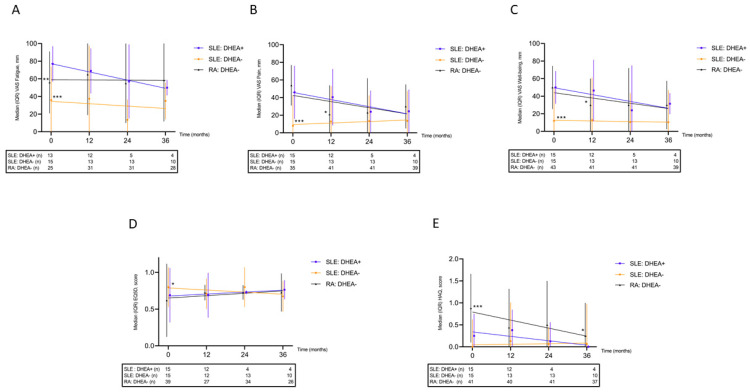
Longitudinal patient-reported outcome measures shown for patients with systemic lupus erythematosus (SLE) exposed/unexposed to DHEA and for sex- and age-matched controls with rheumatoid arthritis (RA); (**A**) visual analogue scale (VAS) fatigue, worse in DHEA-exposed SLE than in SLE/RA controls; (**B**) VAS pain, worse in DHEA-exposed SLE compared with SLE controls; (**C**) VAS well-being, worse in DHEA-exposed SLE compared with SLE controls; (**D**) EQ-5D, worse in DHEA-exposed SLE compared with SLE controls; (**E**) Health Assessment Questionnaire (HAQ), worse in RA compared with both SLE groups. * *p* < 0.05, ** *p* < 0.01, *** *p* < 0.005.

**Figure 3 jcm-11-05300-f003:**
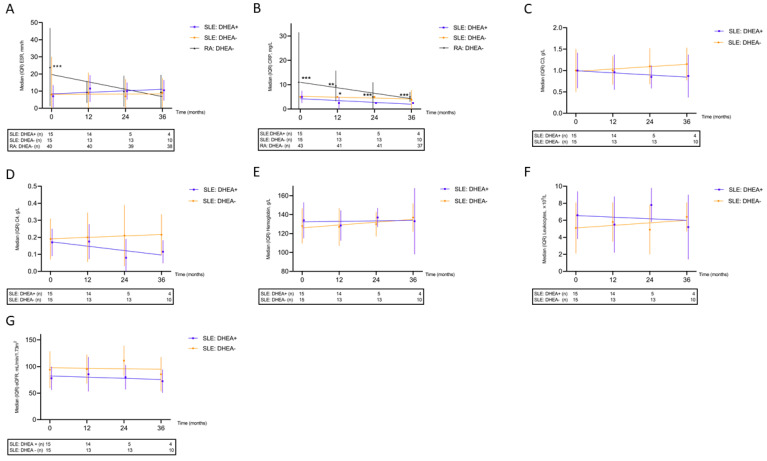
Longitudinal laboratory data demonstrated for patients with systemic lupus erythematosus (SLE) exposed/unexposed to DHEA and for sex- and age-matched controls with rheumatoid arthritis (RA); (**A**) erythrocyte sedimentation rate (ESR), higher in RA compared with both SLE groups; (**B**) C-reactive protein (CRP), higher in RA compared with DHEA-exposed SLE and higher among SLE controls than in DHEA-exposed SLE at 12 months; (**C**) Complement protein 3 (C3); (**D**) Complement protein 4 (C4); (**E**) Hemoglobin concentration; (**F**) Leukocyte count; (**G**) Estimated glomerular filtration rate (eGFR). * *p* < 0.05, ** *p* < 0.01, *** *p* < 0.005.

**Table 1 jcm-11-05300-t001:** Characteristics of the included patients.

Background Characteristics	*Median Value (Range) [IQR] or %*		
	**SLE: DHEA+ (*n* = 15)**	**SLE: DHEA− (*n* = 15)**	**RA: DHEA− (*n* = 45)**
Females, *n* (%)	14 (93.3)	14 (93.3)	42 (93.3)
Caucasian ethnicity, *n* (%)	14 (93.3)	14 (93.3)	N/A
Age at disease onset (years)	42 (12–76) [20]	43 (15–55) [21]	49 (20–76) [19]
Age at baseline (years)	51 (24–76) [19.5]	46 (21–71) [13.5]	50 (21–76) [19]
Disease duration at baseline (years)	9 (0–31) [17]	4 (0–19) [8.5]	1 (0–5) [1]
SLEDAI-2K (score)	0 (0–4) [1]	2 (0–15) [4]	N/A
Physician’s global assessment (0–4)	0 (0–1) [0]	0 (0–2) [0]	N/A
BMI at baseline (kg/m^2^)	27.6 (19.2–40.4) [5.2]	24.5 (19.9–35.8) [6]	N/A
SDI at baseline (score)	0 (0–2) [1]	0 (0–4) [1]	N/A
ACR criteria fulfilled, *n*	5 (3–7)	4 (3–7)	N/A
**1982 ACR criteria, *n* (%)**			
Malar rash	7 (46.7)	8 (53.3)	N/A
Discoid rash	12 (80)	7 (46.7)	N/A
Photosensitivity	1 (6.7)	1 (6.7)	N/A
Oral ulcers	3 (20)	0 (0)	N/A
Arthritis	11 (73.3)	13 (86.7)	N/A
Serositis	4 (26.7)	5 (33.3)	N/A
Renal disorder	6 (40)	6 (40)	N/A
Neurological disorder	1 (6.7)	1 (6.7)	N/A
Hematological disorder	9 (60)	6 (40)	N/A
Immunological disorder	7 (46.7)	8 (53.3)	N/A
Anti-nuclear antibody	15 (100)	15 (100)	N/A

ACR, American College of Rheumatology; BMI, body-mass index; DHEA, dehydroepiandrosterone; n.s., not significant; N/A, not applicable; RA, rheumatoid arthritis; SDI, Systemic Lupus International Collaborating Clinics (SLICC)/ACR damage index; SLE, systemic lupus erythematosus; SLEDAI-2K, systemic lupus erythematosus disease activity index 2000.

**Table 2 jcm-11-05300-t002:** Individual descriptions of the 15 patients with SLE exposed to dehydroepiandrosterone.

Sex	Age at Start (years)	DHEA Exposure (months)	DHEA Concentration, Baseline (μmol/L)	DHEA Concentration, Percent of Lower Reference Limit (%)	Initial Daily DHEA Dose (mg)	Concomitant DMARDs	Steroid Dose at DHEA Initiation (mg)	Change in Steroid Dose at Last Follow Up (mg)	Cause of Cessation
F	57	8	0.38	75	50	HCQ	0	0	Treatment ongoing
F	47	4	N/A	N/A	50	MMF	7.5	0	Without specification *
F	54	4	0.55	57	200	HCQ	0	0	Lack of efficacy *
F	50	6	0.38	40	50	HCQ	2.5	0	Lack of efficacy
M	43	81	N/A	N/A	50	HCQ, MTX	5	+2.5	Treatment ongoing
F	56	3	N/A	N/A	50	None	7.5	−2.5	Acne, scaly hair *
F	31	12	0.54	20	25	HCQ, AZA	2.5	0	Lack of efficacy
F	37	14	2.2	140	25	HCQ	0	0	Without specification
F	27	30	2.7	100	25	HCQ, MMF	2.5	0	Treatment ongoing
F	61	9	0.22	43	50	AZA	5	0	Lack of efficacy
F	47	69	0.35	36	25	HCQ	0	0	Acne, fear of thrombosis
F	58	17	0.44	86	50	HCQ, MMF	5	−5	Treatment ongoing
F	76	36	0.14	42	200	HCQ	5	+2.5	Treatment ongoing
F	23	16	2.7	68	50	HCQ	5	−5	Treatment ongoing
F	50	10	0.43	45	25	HCQ	5	0	Lack of efficacy

* Early cessation (≤4 months). AZA, azathioprine; DHEA, dehydroepiandrosterone; DMARDs, disease-modifying anti-rheumatic drugs; HCQ, hydroxychloroquine; MMF, mycophenolate mofetil; N/A, not applicable.

## Data Availability

Data available on request from the authors.

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
