# Peer review of "Fighting Fatigue in Systemic Lupus Erythematosus: Experience of Dehydroepiandrosterone on Clinical Parameters and Patient-Reported Outcomes"

_jcm, 2022, doi:10.3390/jcm11185300_

Round 1

Reviewer 1 Report

·         The repeated mention of a 36-month-long study throughout the manuscript is misleading. The study duration was 22 years and the mean follow-up was 36 months (± XY Months). E.g. under section 2.1 third paragraph “The patients were followed up to 36 months” which is inconsistent with the duration of follow-up mentioned elsewhere.

·         Please provide the rationale for including RA patients as a comparator group.

·         Please add in methods of how was the analysis adjusted for the censored data.

·         Some subjects have a follow-up duration that is significantly more than 36 months. How and which 36-month interval was chosen for these patients

·         How many patients fulfilled only SLICC SLE criteria vs only ACR SLE criteria? From results section 3.1 it appears that all subjects fulfilled ACR criteria at the baseline. Please confirm and specify in results.

·         There is a significantly higher drop in the number of DHEA+ subjects during follow-up as compared to DHEA- subjects. Please provide an explanation for this difference.

·         Most subjects (9/15) on DHEA discontinued either due to either lack of efficacy or treatment-related side effects.  This point needs to be mentioned under section 3.4. In addition, please rephrase the 2nd sentence of the second paragraph under section 3.4 to more accurately describe DHEA use in the 15 patients.

·         The results showing a comparison of PROMs between three groups is problematic since there is unequal censoring of longitudinal data since there is no longitudinal data from a large number of DHEA+ SLE patients as compared to other groups. This should be clearly mentioned in the results and discussion as a major limitation.

·         Results section 3.6 the first sentence needs to be removed or rephrased since this was an unblinded study and there is confounding by indication.

·         The second paragraph under discussion has statements such as “… DHEA to patients with SLE is generally well tolerated” and “…DHEA-treated group did not accumulate more organ damage..” which are inconsistent with the results.

·         There is a typo on line 310 under discussion: “same” =” some”

·         The conclusions should be toned down in reference to DHEA being well tolerated and that “data indicate” benefit of DHEA on improving fatigue. 

Author Response

Comments and Suggestions for Authors

  • The repeated mention of a 36-month-long study throughout the manuscript is misleading. The study duration was 22 years and the mean follow-up was 36 months (± XY Months). E.g. under section 2.1 third paragraph “The patients were followed up to 36 months” which is inconsistent with the duration of follow-up mentioned elsewhere.

Response: Thank you for the careful review of our manuscript. Regarding the length of the study, in fact all included SLE patients were part of a clinical research and quality register with longitudinal follow-up (Clinical Lupus Register in North-eastern Gothia; Swedish acronym KLURING) [Heijke R, Ahmad A, Frodlund M, Wirestam L, Dahlström Ö, Dahle C, Kechagias S, Sjöwall C. Usefulness of Clinical and Laboratory Criteria for Diagnosing Autoimmune Liver Disease among Patients with Systemic Lupus Erythematosus: An Observational Study. J Clin Med. 2021 Aug 26;10(17):3820. doi: 10.3390/jcm10173820. PMID: 34501268; PMCID: PMC8432263]. The current study had a retrospective and observational 36-month study design where all patients in KLURING exposed to DHEA were matched to SLE comparators from the KLURING cohort unexposed to DHEA. We understand that this could be confusing and apologize for the misunderstanding. In the revised version, we have tried to clarify this further.

  • Please provide the rationale for including RA patients as a comparator group.

Response: This is a relevant question. The SLE subjects receiving DHEA were not the most somatically ill patients in KLURING as illustrated in Table 1 by the limited degree of accrued damage (SDI). Instead, they were fatigued, and the majority had mild low-to-moderately active disease with particularly joint or musculoskeletal involvement as illustrated by the low SLEDAI-2K scores in Table 1. Since the SLE comparators (DHEA-) were slightly different – not statistically significant though – in terms of disease activity and organ involvement, we decided to also include age- and sex-matched patients with recent-onset rheumatoid arthritis (RA). Apparently, all patients with early RA had signs of active joint involvement. Consequently, we consider this group as a valuable comparator since organ involvement was similar to SLE DHEA+ patients and the PROMs data for RA subjects had been collected in the same way as for the SLE patients. We have further clarified the rationale for including RA patients in the revised version.

  • Please add in methods of how was the analysis adjusted for the censored data.

Response: No adjustments of the data were made. This information has been added to the Statistics (2.4).

  • Some subjects have a follow-up duration that is significantly more than 36 months. How and which 36-month interval was chosen for these patients

Response: Yes, that is correct. As the dropout increased after 36 months, we decided to only provide detailed information on the first 36 months but indeed some patients obviously benefited from DHEA and remained on the drug much longer (Table 2). These patients were seen regularly by rheumatologist at our clinic, at least annually.

  • How many patients fulfilled only SLICC SLE criteria vs only ACR SLE criteria? From results section 3.1 it appears that all subjects fulfilled ACR criteria at the baseline. Please confirm and specify in results.

Response: This is a relevant question. In Table 1, the range of “ACR criteria fulfilled, n” shows that some patients only fulfilled 3 ACR criteria (≥4 is required for SLE classification according to ACR-82). However, all SLE patients (both DHEA+ and DHEA-) meeting less than 4 ACR criteria fulfilled the SLICC criteria set [Ighe A, Dahlström Ö, Skogh T, Sjöwall C. Application of the 2012 Systemic Lupus International Collaborating Clinics classification criteria to patients in a regional Swedish systemic lupus erythematosus register. Arthritis Res Ther. 2015 Jan 10;17(1):3. doi: 10.1186/s13075-015-0521-9. PMID: 25575961; PMCID: PMC4318183]. In the DHEA+ group, 2/15 met 3 ACR criteria; and in the DHEA- group, 3/15 met 3 ACR criteria. The response to DHEA was not different among patients fulfilling vs. not fulfilling ACR-82 criteria. This information has been added in the revised version.

  • There is a significantly higher drop in the number of DHEA+ subjects during follow-up as compared to DHEA- subjects. Please provide an explanation for this difference.

Response: Yes, it is true that the dropout was higher in the DHEA+ group. This was mainly due to side-effects, or lack of efficacy, as demonstrated in Table 2. We have further clarified this in the revised version.

  • Most subjects (9/15) on DHEA discontinued either due to either lack of efficacy or treatment-related side effects.  This point needs to be mentioned under section 3.4. In addition, please rephrase the 2ndsentence of the second paragraph under section 3.4 to more accurately describe DHEA use in the 15 patients.

Response: Yes, it is true (see previous comment). This information has now been added to section 3.4. and the sentence rephrased as suggested by the Reviewer.

  • The results showing a comparison of PROMs between three groups is problematic since there is unequal censoring of longitudinal data since there is no longitudinal data from a large number of DHEA+ SLE patients as compared to other groups. This should be clearly mentioned in the results and discussion as a major limitation.

Response: We admit that this is a limitation, and this has been further acknowledged in the Discussion. Still, as it has not previously been reported, we believe that observational real-life experience of DHEA in SLE is important to communicate. We figured that it was a better way to report observational data with comparator groups than without.

  • Results section 3.6 the first sentence needs to be removed or rephrased since this was an unblinded study and there is confounding by indication.

Response: This is well taken. The sentence has been rephrased. Nota bene, at baseline (month 0), none of the groups had been exposed to DHEA.

  • The second paragraph under discussion has statements such as “… DHEA to patients with SLE is generally well tolerated” and “…DHEA-treated group did not accumulate more organ damage..” which are inconsistent with the results.

Response: We have rephrased the first sentence as suggested by the Reviewer. However, we demonstrate in 3.2. that accumulation of organ damage in the two SLE groups over the 36 months were not significantly different.  

  • There is a typo on line 310 under discussion: “same” =” some”

Response: Thank you. This has been corrected.

  • The conclusions should be toned down in reference to DHEA being well tolerated and that “data indicate” benefit of DHEA on improving fatigue. 

Response: We agree, and the conclusions have been toned down according to the suggestions made by the Reviewer.

Reviewer 2 Report

The manuscript "Fighting Fatigue in Systemic Lupus Erythematosus: Experience of Dehydroepiandrosterone on Clinical Parameters and Patient-Reported Outcomes" aimed to illustrate the value of DHEA supplementation in a predominantly DHEA-deficient SLE cohort. The subject of the work is interesting and relevant to both researchers and clinicians. However, a few issues are to be addressed by the authors:

·        Table 1 presents some of your results, it should not be present in the "Methods" section.

·        Table 1 states that at least 2 of your patients were over 70 years old (76 and 71) when they were diagnosed with SLE (age at onset). How did you exclude a paraneoplastic etiology of their symptoms? Please provide a detailed description and a diagram of the inclusion and exclusion processes.

·        What was the p value for the median age in SLE DHEA+ versus SLE DHEA- patients? What test did you apply to analyze the data in Table 1?

·        L123-124: "patient's height and length" - "weight"

·        "to assess general QoL (0=dead, 1=perfect QoL)" = ?

·        You state that you used Euro-Qol 5 D to assess quality of life. Please provide a more detailed description of the patients' evolution in this respect (a description of each of the 5 dimensions). Also, please provide a more detailed description of your results regarding HAQ.

In its present version, the manuscript does not contain sufficient information. Moreover, its retrospective nature is a major limitation.

Author Response

Comments and Suggestions for Authors

The manuscript "Fighting Fatigue in Systemic Lupus Erythematosus: Experience of Dehydroepiandrosterone on Clinical Parameters and Patient-Reported Outcomes" aimed to illustrate the value of DHEA supplementation in a predominantly DHEA-deficient SLE cohort. The subject of the work is interesting and relevant to both researchers and clinicians. However, a few issues are to be addressed by the authors:

  • Table 1 presents some of your results, it should not be present in the "Methods" section.

Response: Thank you for the careful review of our manuscript. The majority of the information in Table 1 is baseline characteristics. However, as the two comparator groups were sex- and age-matched to the SLE DHEA+ group, it was reasonable to also demonstrate potential differences between the groups. We acknowledge that the latter is related to Results (Column for p-values). Consequently, we removed this column from Table 1 and kept the table in the Methods section. 

  • Table 1 states that at least 2 of your patients were over 70 years old (76 and 71) when they were diagnosed with SLE (age at onset). How did you exclude a paraneoplastic etiology of their symptoms? Please provide a detailed description and a diagram of the inclusion and exclusion processes.

Response: This is a relevant question. In fact, in the DHEA+ group there was one female patient with SLE at the age of 76 based on arthritis, skin rash, complement consumption, positive ANA, anti-dsDNA and anti-SSA/Ro52 antibodies. This lady was diagnosed with SLE in 2017. She is doing well and has now been followed by us at the Rheumatology unit during five years, which makes paraneoplastic syndrome very unlikely. We admit that the age of SLE debut for the patient is somewhat unusual, but SLE onset later in life is not unique in Scandinavia [Hermansen ML, Lindhardsen J, Torp-Pedersen C, Faurschou M, Jacobsen S. Incidence of Systemic Lupus Erythematosus and Lupus Nephritis in Denmark: A Nationwide Cohort Study. J Rheumatol. 2016 Jul;43(7):1335-9. doi: 10.3899/jrheum.151221. Epub 2016 May 1. PMID: 27134247].

Regarding the 71-year-old patient, this is probably a misunderstanding, as this refers to “Age at baseline” and not to “Age at disease onset” (see the different lines in Table 1).

Regarding the inclusion and exclusion processes, all included SLE patients were part of a clinical research and quality register with longitudinal follow-up (Clinical Lupus Register in North-eastern Gothia; Swedish acronym KLURING) [Heijke R, Ahmad A, Frodlund M, Wirestam L, Dahlström Ö, Dahle C, Kechagias S, Sjöwall C. Usefulness of Clinical and Laboratory Criteria for Diagnosing Autoimmune Liver Disease among Patients with Systemic Lupus Erythematosus: An Observational Study. J Clin Med. 2021 Aug 26;10(17):3820. doi: 10.3390/jcm10173820. PMID: 34501268; PMCID: PMC8432263]. All included SLE patients fulfilled the 1982 American College of Rheumatology (ACR-82) and/or the 2012 Systemic Lupus International Collaborating Clinics (SLICC) classification criteria [Ighe A, Dahlström Ö, Skogh T, Sjöwall C. Application of the 2012 Systemic Lupus International Collaborating Clinics classification criteria to patients in a regional Swedish systemic lupus erythematosus register. Arthritis Res Ther. 2015 Jan 10;17(1):3. doi: 10.1186/s13075-015-0521-9. PMID: 25575961; PMCID: PMC4318183].

The current study is a retrospective and observational 36-months study design where all patients in KLURING exposed to DHEA were matched to SLE comparators unexposed to DHEA. We prefer to refer to our previous publications of KLURING, instead of constructing a diagram. In the DHEA+ group, 2/15 met SLICC criteria in the absence of ACR-82; and in the DHEA- group, 3/15 met SLICC criteria only. The response to DHEA was not different among patients fulfilling vs. not fulfilling the ACR-82 criteria. We have further clarified this in the revised version.

  • What was the p value for the median age in SLE DHEA+ versus SLE DHEA- patients? What test did you apply to analyze the data in Table 1?

Response: The p-value of the median age between the two SLE-groups was 0.389 and did not meet statistical significance. For analysis of the three groups, initially Kruskal-Wallis test was applied. If the result reached statistical significance (p<0.05), a confirmatory Mann-Whitney U test was applied to determine between which of the groups there was statistical significance. For determining significance between fulfilled ACR criteria, Chi-2 testing was applied. This information is given in 2.4.

  • L123-124: "patient's height and length" - "weight"

Response: Thank you. This has been corrected.

  • "to assess general QoL (0=dead, 1=perfect QoL)" = ?

Response: The Swedish version of the EQ-5D-3L was used to assess general health based on five different dimensions in order to derive a utility index which provides a score indexed at 1 (perfect health) and 0 (dead) [Dolan P. Modeling valuations for EuroQol health states. Med Care. 1997 Nov;35(11):1095-108. doi: 10.1097/00005650-199711000-00002. PMID: 9366889]. We have further clarified this in the revised version.

  • You state that you used Euro-Qol 5 D to assess quality of life. Please provide a more detailed description of the patients' evolution in this respect (a description of each of the 5 dimensions). Also, please provide a more detailed description of your results regarding HAQ.

Response: The EQ-5D is a well-established instrument to assess QoL [Dolan P. Modeling valuations for EuroQol health states. Med Care. 1997 Nov;35(11):1095-108. doi: 10.1097/00005650-199711000-00002. PMID: 9366889]. The EQ-5D-3L descriptive system comprises the following five dimensions: mobility, self-care, usual activities, pain/discomfort and anxiety/depression. Each dimension has 3 levels: no problems, some problems, and extreme problems. The patient is asked to indicate his/her health state by ticking the box next to the most appropriate statement in each of the five dimensions. This decision results into a 1-digit number that expresses the level selected for that dimension. The digits for the five dimensions can be combined into a 5-digit number that describes the patient’s health state. We have further clarified this in the revised version. The Health Assessment Questionnaire (HAQ) is a validated patient-reported tool to assess presence of activity limitations in RA but it has also been used in SLE [Heijke R, Björk M, Thyberg I, Kastbom A, McDonald L, Sjöwall C. Comparing longitudinal patient-reported outcome measures between Swedish patients with recent-onset systemic lupus erythematosus and early rheumatoid arthritis. Clin Rheumatol. 2022 May;41(5):1561-1568. doi: 10.1007/s10067-021-05982-3. Epub 2021 Nov 27. PMID: 34839415; PMCID: PMC9056441]. We have added further clarifications in the revised version and the results regarding EQ-5D and HAQ were discussed.

In its present version, the manuscript does not contain sufficient information. Moreover, its retrospective nature is a major limitation.

Response: With the added information in the revised version, we hope that the manuscript is more comprehensive and addresses the concerns raised by the Reviewer. We admit that the retrospective study design is a limitation, and this has been further acknowledged in the Discussion. Still, as it has not previously been reported, we believe that observational real-life experience of DHEA in SLE is important to communicate.

Round 2

Reviewer 1 Report

Thank you for addressing the comments. No further comments or suggestions.  

Reviewer 2 Report

The authors have addressed my concerns. I have no further comments.